# Evolutionary Analysis of the Land Plant-Specific TCP Interactor Containing EAR Motif Protein (TIE) Family of Transcriptional Corepressors

**DOI:** 10.3390/plants14152423

**Published:** 2025-08-05

**Authors:** Agustín Arce, Camila Schild, Delfina Maslein, Leandro Lucero

**Affiliations:** 1Instituto de Agrobiotecnología del Litoral (CONICET-UNL), Universidad Nacional del Litoral, Colectora Ruta Nacional 168 km 0, Santa Fe 3000, Argentina; ial@santafe-conicet.gov.ar (A.A.); camiluschild@gmail.com (C.S.); delfinamaslein@hotmail.com (D.M.); 2Facultad de Bioquímica y Ciencias Biológicas, Universidad Nacional del Litoral, Colectora Ruta Nacional 168 km 0, Santa Fe 3000, Argentina; 3Facultad de Humanidades y Ciencias, Universidad Nacional del Litoral, Colectora Ruta Nacional 168 km 0, Santa Fe 3000, Argentina

**Keywords:** TCP Interactor containing EAR motif protein (TIE), TEOSINTE BRANCHED1/CYCLOIDEA/PCF (TCP), evolution, gene co-expression, embryophyte

## Abstract

The plant-specific TCP transcription factor family originated before the emergence of land plants. However, the timing of the appearance of their specific transcriptional repressor family, the TCP Interactor containing EAR motif protein (TIE), remains unknown. Here, through phylogenetic analyses, we traced the origin of the TIE family to the early evolution of the embryophyte, while an earlier diversification in algae cannot be ruled out. Strikingly, we found that the number of TIE members is highly constrained compared to the expansion of TCPs in angiosperms. We used co-expression data to identify potential TIE-TCP regulatory targets across *Arabidopsis thaliana* and rice. Notably, the expression pattern between these species is remarkably similar. TCP Class I and Class II genes formed two distinct clusters, and TIE genes cluster within the TCP Class I group. This study provides a comprehensive evolutionary analysis of the TIE family, shedding light on its conserved role in the regulation of gene transcription in flowering plant development.

## 1. Introduction

The plant-specific TEOSINTE BRANCHED1/CYCLOIDEA/PCF (TCP) family of transcription factors (TFs) was named after its founding members: TEOSINTE BRANCHED1 (TB1) in maize (*Zea mays*), CYCLOIDEA (CYC) in *Antirrhinum majus*, and PROLIFERATING CELL FACTORS 1 and 2 (PCF1/2) in rice (*Oryza sativa*) [1,2,3,4]. The TCP family predates the origin of land plants and is phylogenetically divided into two main clades: Class I (TCP-P) and Class II (TCP-C). These TFs play critical roles in diverse developmental and physiological processes, including seed germination, root and shoot architecture, leaf morphogenesis, and floral organ development [5,6]. Intriguingly, TCPs emerged prior to plant terrestrialization, with both classes already present in charophyte algae [7]. However, the family underwent significant expansion during land plant evolution. For instance, the alga *Chara braunii* possesses only two TCPs (one Class I and one Class II), whereas flowering plants like *Arabidopsis thaliana* encode 24 distinct TCPs [4,7,8]. TCP TFs bind DNA to regulate gene expression, often exhibiting functional redundancy or antagonism across model and crop species. Genome-wide studies in Arabidopsis and maize suggest that TCPs primarily act as transcriptional activators [9,10,11]. TCP Class I and II recognize related but different cis elements, GTGGGNCC and GTGGNCCC, respectively [12,13,14]. More recently, the TCP domain structure was resolved, revealing a fold composed of two α-helix and a β-strand within [15].

Given the importance of TCP transcription factors in plant development and evolution, the study of the molecular mechanisms regulating their activity—as key regulators of gene transcription—has received significant attention in recent decades. A small, plant-specific family of transcriptional regulators, called TCP Interactor containing EAR motif proteins (TIEs), was identified in *A. thaliana*. These proteins interact with a significant subset of TCP transcription factors and repress their transcriptional activity [16]. TIE proteins are characterized by an N-terminal helical region and a C-terminal EAR motif (DLELRL). In *A. thaliana*, TIE1 (AtTIE1) interacts with at least eight different TCP proteins and, through its EAR motif, recruits the transcriptional corepressors TOPLESS (TPL) and TOPLESS-RELATED (TPR) proteins. Notably, normal leaf development in Arabidopsis depends on this molecular hub [16]. Additionally, branch development is modulated by the interaction between TIE1 and the TCP master regulator of axillary bud dormancy, BRANCHED 1 (BRC1/TCP18). AtTIE1 impairs BRC1-mediated activation of the HD-ZIP transcription factor HB21, thereby promoting axillary bud dormancy release [17]. Strikingly, this regulatory mechanism is evolutionarily conserved. In cotton (*Gossypium hirsutum*), GhTIE1 interacts with GhBRC1 to suppress *GhHB21* transcription, inducing axillary branch development [18]. More recently, AtTIE1 and AtTIE2 were shown to regulate cytokinin responses, modulating root development in Arabidopsis. Furthermore, TIE2 interacts with type-B ARR1, acting as a transcriptional corepressor through its EAR motif [19]. In rice (*Oryza sativa*), the only Monocot species where the molecular and biological roles of a TIE protein have been studied to date, OsTIE1 interacts with TCP transcription factors, functioning as a transcriptional corepressor. Notably, the OsTIE1-OsTCP1 module controls anther dehiscence and male sterility by directly regulating jasmonic acid (JA) biosynthesis, suggesting a potential target for hybrid rice breeding [20]. Collectively, these findings underscore the diverse and conserved roles of TIE proteins across plant species. Given their ability to repress TCP transcriptional activity, further research is needed to elucidate their interaction networks, regulatory impacts, and evolutionary significance across the green lineage.

Thus, here we established a comprehensive phylogenetic framework that will be crucial for guiding future research on the TIE family. We traced the origin of the TIE family to the last common ancestor of embryophytes. Within the TIE family, two main clades were identified in angiosperms, Clade A and Clade B, with Clade A being ancestral and Clade B derived. Furthermore, Clade B emerged prior to the divergence between Eudicots and Monocots, coinciding with early angiosperm diversification. Notably, TIE protein underwent limited diversification within angiosperms, compared to its TCP TF targets. Protein motif analyses not only supported the phylogenetic topology but also revealed clade-specific domains critical for major flowering plant lineages, as well as distinct features differentiating TIE Clade A and Clade B. Notably, co-expression analyses of *TIE* and *TCP* genes in Arabidopsis and rice revealed a conserved pattern of expression, suggesting common regulatory modes of action within flowering plants.

## 2. Results and Discussion

### 2.1. Two Alternative Scenarios Might Explain the Origin of TIE Proteins

BLAST searches using *A. thaliana* TIE proteins as queries were conducted. We searched the Phytozome (https://phytozome-next.jgi.doe.gov/; accessed on 23 November 2024) and Bryogenome portal (https://bryogenomes.org/; accessed on 28 November 2024) databases to cover genomes within viridiplantae. No significant hits were obtained for algal species. Since algal genomes are underrepresented in these databases, we expanded the search to other algal genomes using the Algal Genomics Resource database (https://mycocosm.jgi.doe.gov/mycocosm/home; accessed on 2 December 2024), yet no hits were found. To further rule out the presence of TIE protein in algae, we performed additional BLAST analyses modifying parameters of the search (see Section 3.1), also without positive hits. However, significant hits were restricted to sequences from species of the embryophyte clade. TIE proteins were identified in mosses (*Physcomitrium patens*, *Sphagnum fallax*, *Ceratodon purpureus*) and the liverwort *Marchantia polymorpha*. Given that mosses and liverworts are sister groups [21], TIE proteins likely originated within or before the last common ancestor of embryophytes. A previous phylogenetic analysis placed the root of the TIE protein family within mosses [22]. Here, by including *M. polymorpha*, we demonstrate that the origin of TIE proteins may be even earlier. To further test for potential algal TIE homologs, we performed an additional BLAST search using the identified *M. polymorpha* TIE sequence as a query against the aforementioned algae genome databases. Again, no algal hits were recovered, reinforcing the possibility that TIE proteins are absent in currently available algal genomes. However, since TIE proteins are present in all embryophytes, this might suggest that the ancestor of land plants may have possessed TIE proteins, or even that their origin predates the emergence of land plants. Thus, although no TIE proteins have been identified in algae yet, we speculate that TIE proteins may have originated within the algal lineage. Given the ubiquitous presence of TIE proteins across embryophytes, two alternative evolutionary scenarios emerge: (1) the ancestor of all land plants already possessed TIE proteins, or (2) their origin predates land plant emergence, possibly within an algal lineage that remains unsampled or extinct. Resolving this ambiguity will require expanded genomic sampling from undersampled algal groups.

### 2.2. Diversification Trends of TIE Proteins Contrast with TCP Expansion in Angiosperms

The fully resolved tree topology obtained through Maximum Likelihood (ML) of TIE proteins reveals a distinct diversification pattern compared to their targets, the TCP transcription factors. ML trees show that basal embryophyte TIE proteins form a cohesive cluster with strong branch support, as do those from basal angiosperms. Two well-supported clades emerged within flowering plants, TIE Clade A and B. TIE Clade A is basal to the family and includes *A. thaliana* TIE1/TIE2 and basal angiosperm sequences. TIE Clade B exclusively comprises Monocot and Eudicot sequences and includes AtTIE3/TIE4. In addition, angiosperm TIE sequences resolve into Monocot and Eudicot clades, confirming deep divergence within flowering plants (Figure 1).

We observed that some branches exhibit high specific variation as indicated by their branch length (Figure 1). Therefore, to test for long-branch attraction (a common phylogenetic artifact), we reconstructed an angiosperm-only ML tree rooted with *Amborella trichopoda*, the sister lineage to flowering plants [23]. This analysis retained the TIE Clade A/B division, reinforcing its validity. Again, Monocot and Eudicot clades were also recovered (Figure 2).

One of the most striking findings from our phylogenetic analyses is the limited number of TIE members per species. TIE proteins underwent limited diversification within angiosperms, where most Monocot and Eudicot species contain only three to four TIE copies (Figure 1 and Figure 2). This stands in sharp contrast to the dramatic expansion observed in their target TCP transcription factors, which increased from 2 members in *M. polymorpha* to over 20 in most angiosperms [4,24]. This limited diversification is evident in both diploid and polyploid species. For example, Arabidopsis has 4 TIE and 24 TCP copies, while the polyploid cotton (*Gossypium hirsutum*) has 4 and 74 [25], respectively. Together, these results suggest that TIE proteins evolved under a constrained diversification pattern compared to TCPs, possibly reflecting functional or regulatory implications.

Notably, protein–protein interaction screens have shown that AtTIE1 interacts with all eight tested members of the CIN-like TCP transcription factors [16]. Moreover, TIE1 represses BRC1/TCP18-direct target genes through direct interaction, a mechanism conserved in both Arabidopsis and cotton [17,18]. These findings demonstrate that a single TIE protein can interact with—and likely repress the transcriptional activity of—at least nine TCP TFs in Arabidopsis. This suggests that TIE-mediated repression of TCP function remains effective despite sequence variation among TCP proteins, likely through interaction with the conserved TCP domain. Therefore, the limited diversification of TIE proteins might be sufficient to regulate most TCPs, a molecular mechanism that appears to be determined by the co-expression patterns of *TIE* and *TCP* genes.

### 2.3. Diversification of Protein Motifs Supports TIE Family Classification and Reveals Potential Neo-Functionalization

To explore the evolutionary variability of TIE proteins, we identified conserved motifs using the MEME suite (https://meme-suite.org/meme/; accessed on 5 February 2025). Three motifs were previously characterized as hallmarks of the TIE family: a helical region, an EAR repressor motif (DLELRL), and a nuclear localization signal (KRGK) [16]. Our analysis confirmed Motifs 1 and 2 as the most conserved features across embryophytes (Figure 3 and Figure 4). Motif 1, which includes the helical region, is highly conserved in both sequence and position near the N-terminus. A truncated N-terminus version of the AtTIE1 repressor containing Motif 1 was sufficient to interact with BRC1 in a yeast-two-hybrid experiment [18]. Notably, Motif 1 is present in all TIE members across embryophytes (Figure 3), and it is also the most conserved domain at the sequence level (Figure 4). This suggests that the conservation of this domain is crucial to modulate and preserve the protein interaction network across evolution. Motif 2 harbors the EAR repressor domain at the C-terminus, mediating interaction with the TOPLESS repressor [16], suggesting that all TIE proteins likely retain transcriptional repressive activity against their TCP TF targets. Disruption of Motif 2 of *A. thaliana* TIE1 resulted in the loss of the interaction with TOPLESS. Hence, TCP activity repression exerted by TIE proteins is abolished when Motif 2 is mutated [16]. The most conserved part of Motif 2 is the C-terminus containing the LxLxL. This suggests that selection acted to conserve the Leucine residues crucial to recruit TOPLESS proteins (Figure 4). Considering that Motif 1 and Motif 2 are responsible for interaction with TCP and TOPLESS proteins, respectively, the activity of TIE proteins as a co-repressor of TCP activity might have been acquired earlier during the origin of land plants.

The nuclear localization signal (NLS) is embedded within Motif 8, though its presence is not universal. While this motif originated in basal embryophytes, it is absent in the moss *Selaginella moellendorffii* and entirely missing in Clade A for angiosperm species. Strikingly, it reappears in Clade B proteins (e.g., AtTIE3 and AtTIE4), suggesting evolutionary reversion. Given that TIE1 localizes to nuclei and interacts with nuclear proteins [16,19], alternative mechanisms for nuclear import or functional divergence may exist that are worth exploring in the future.

Beyond these conserved motifs, we identified 12 novel motifs, further supporting the phylogenetic division of TIE proteins into Clade A and Clade B. Motifs 3–5, 7, and 13 are exclusive to Clade A, while Motifs 8, 10, and 15 are specific to Clade B. Although Motifs 1 and 2 are shared, another clade-specific motif suggests functional diversification, particularly in Clade B. Notably, Motifs 3–6, 8–9, and 11–14 are absent in Monocots, highlighting divergent evolution of TIE proteins in this lineage compared to other angiosperms. The function of these novel motifs is yet enigmatic for TIE proteins. Only Motifs 1, 2, and 8 were previously linked to TIE function in *A. thaliana*, rice, and cotton [16,17,18].

### 2.4. Co-Expression Analyses Revealed Conserved Patterns of TIE and TCP Genes Across Flowering Plants

Leveraging the wealth of available transcriptomic data of Arabidopsis and rice, we explored the expression pattern of both TIE and TCP genes to seek potential conserved patterns between these tightly connected gene families. We used transcriptomic data from the ATTED-II database version 12 [26], which analyzed publicly available RNA-seq data and generated normalized expression levels for each gene in all sequencing experiments and, based on these results, calculated a co-expression index for each gene pair. Cluster analysis of Arabidopsis TCP genes based on these co-expression indices is presented in Figure 4, and, to better assess the relevance of these values, their distribution for TCP and TIE genes with all other genes is presented in density plots (Appendix A).

Co-expression clustering showed a clear distinction in expression patterns between Class I and Class II (Figure 5). Notably, we observed a well-defined cluster comprising Class II TCP2, 3, 4, 10, and 24, which are co-expressed. This finding aligns with the known heterodimer formation among these TCP transcription factors and, more importantly, with their reported functional redundancy [4,6]. Similarly, phylogenetically related Class I TCPs also clustered together. For instance, *TCP14* and *TCP15* exhibited co-expression, consistent with their shared molecular and biological roles [27,28]. While the expression patterns of *TCP* genes were somewhat expected, our analysis also revealed several major trends in *TIE* gene expression. A distinct positive association was observed between Class I *TCPs* and *TIE* gene expression, whereas a negative correlation was found with Class II *TCP* expression (Figure 5). Given that several Class II *TCPs* have been reported to interact with TIE proteins, this pattern may suggest that these TCPs could be released from TIE-mediated modulation through protein–protein interactions, thereby activating their target genes. Another interesting finding was the co-expression of *TIE1* and *TIE2*, which are closely related in our phylogenetic analyses (Figure 1). Moreover, the co-expression of *TIE1* and *TIE2* may also be in agreement with their redundant functions in Arabidopsis root development (19). In general, *AtTIE1* and *AtTIE2* depicted a highly similar expression pattern, opposite to *AtTIE4* or other *TCP* genes. Strikingly, the expression profiles of rice *TIE* and *TCP* genes reveal a conserved co-expression pattern between the two gene families (Figure 6). Notably, *TCP* Class I genes cluster together and are separated from *TCP* Class II. Within the latter, there is a group comprising *OsTCP1*, *2*, *8*, *11*, *14*, and *23*, which are highly co-expressed. Most importantly, *OsTIE1* and *OsTIE2*, closely related evolutionarily (Figure 1), cluster within *TCP* Class I genes. This *TIE* gene expression pattern was similar between rice and Arabidopsis, suggesting a conserved modulation of *TIE* expression in flowering plants. This observation is also somewhat consistent with the conserved function of TIE1 in axillary branch suppression in Arabidopsis and cotton [16,18].

## 3. Materials and Methods

### 3.1. Sequence Retrieval, Phylogenetic Analyses, and Protein Domains Identification

The search for TIE sequences was performed using the BLAST algorithm with default parameters, as implemented in the consulted genome databases, Phytozome v.14 and Phycoscome v1. As a query, we used TIE sequences from *Arabidopsis thaliana* (AGIs). Additionally, to survey basal algal genomes, we used the TIE protein sequence from the basal liverwort *Marchantia polymorpha* as a query. To maximize the chances of finding a distant homolog, we modified different BLAST parameters. We chose the smallest word size available at NCBI, in order to allow for short seed matches that initiate an alignment. We changed the default substitution matrix to a matrix better suited for long evolutionary distances (as BLOSUM45 or PAM250). We also increased the E-value threshold to include all potential hits, although given that it is not extremely short, we do not consider a hit homologous with E-values above 10 × 10^−6^, as it is usually recommended. The sequences obtained are listed in Appendix A. The retrieved protein sequences were aligned using MUSCLE (https://www.ebi.ac.uk/jdispatcher/msa/muscle, accessed on 30 June 2025). To determine the best substitution model for our datasets, we used ModelFinder as implemented in the IQ-TREE 3.0.0 package using the Bayesian Information Criterion to choose the fittest model [29,30]. IQ-TREE was also used to infer the TIE protein evolution under the Maximum Likelihood criterion (Random seed number: 226217) and to estimate the branch support using the UltraFast Bootstrap approach [31]. The resulting Newick tree file was visualized using FigTree v1.4.4 (https://tree.bio.ed.ac.uk/software/figtree/, accessed on 30 June 2025).

To identify the conserved protein domains and their distribution across the protein structures within the TIE family, we utilized the MEME suite (https://meme-suite.org/meme/, accessed on 30 June 2025). Additionally, we examined the spatial arrangement of these domains to assess potential functional implications.

### 3.2. Gene Co-Expression Analyses

To evaluate co-expression of *TCP* and *TIE* genes in *A. thaliana* and rice, we used the ATTED-II database version 12 [26], which contains pairwise co-expression indices for each gene with respect to every other gene in the species. We used the indices based on RNA-Seq transcriptomic data, which include 22,408 transcriptomes (runs) in Arabidopsis and 1149 in rice. These indices were processed in R (R Core Team, 2025) and the heatmaps generated with pheatmap (https://cran.r-project.org/web/packages/pheatmap, accessed on 30 June 2025). For the co-expression index distribution, density plots were produced with packages from the Tidyverse package suite [32].

## 4. Conclusions

Our study reveals that TIE proteins originated in early land plants and diversified into two main clades (A and B), with Clade B emerging before Monocot–Eudicot divergence. Despite the massive expansion of *TCP* genes in angiosperms, TIE proteins remain few (3–4 copies per species), likely due to their ability to broadly repress multiple TCPs via conserved EAR motifs. Co-expression studies in Arabidopsis and rice showed that *TIE* genes associate more strongly with Class I TCPs, indicating conserved regulatory mechanisms. Notably, *TIE1* and *TIE2* from Clade A exhibit similar expression patterns, supporting their redundant roles in development.

Future research should explore TIE functions in non-model species and their potential for crop improvement, particularly in modifying plant architecture. This work provides a foundation for understanding the TIE-TCP regulatory network and its role in plant development and evolution.

## Figures and Tables

**Figure 1 plants-14-02423-f001:**
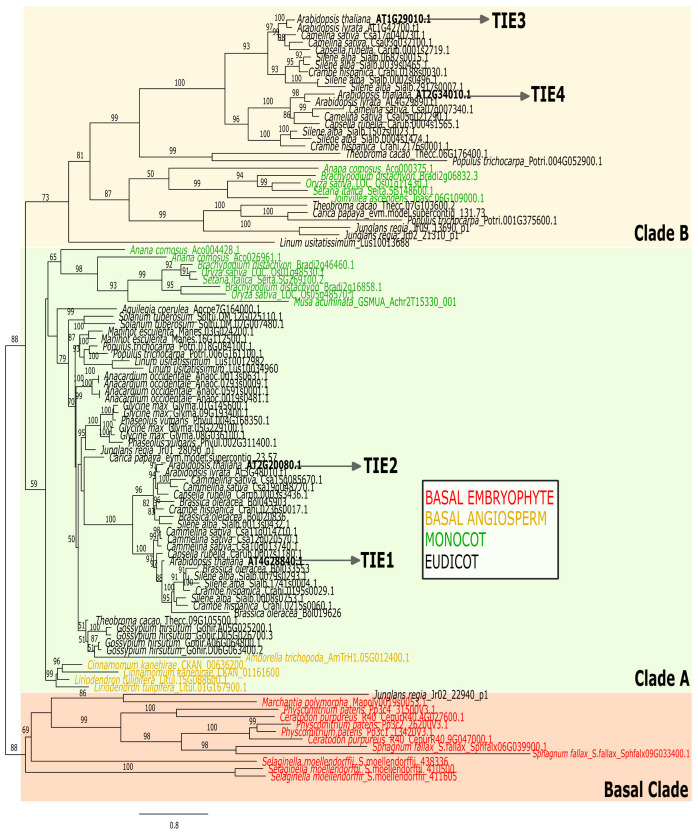
Embryophyte Maximum Likelihood unrooted tree of the TIE protein family. The Log-likelihood of the tree was −43190.277. Arabidopsis thaliana sequences are bolded and marked with an arrow. The sequences of major groups of plants within embryophytes are colored following the label of the white square panel within the graph. The model of substitution determined with ModelFinder was TVMe + I + G4. Values above branches depict bootstrap support (1000 replicates).

**Figure 2 plants-14-02423-f002:**
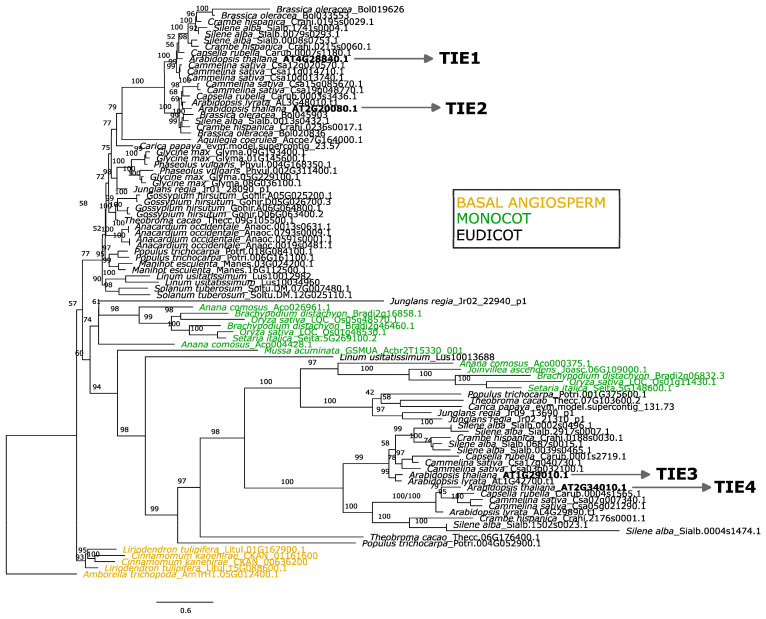
Angiosperm Maximum Likelihood rooted tree of the TIE protein family. The Log-likelihood of the tree was −30803.0625. Arabidopsis thaliana sequences are bolded and marked with an arrow. The sequences of major groups of flowering plants are colored following the label of the square panel within the graph. The model of substitution determined with ModelFinder was JTTDCMut + F + I + G4. Values above branches depict bootstrap support (1000 replicates).

**Figure 3 plants-14-02423-f003:**
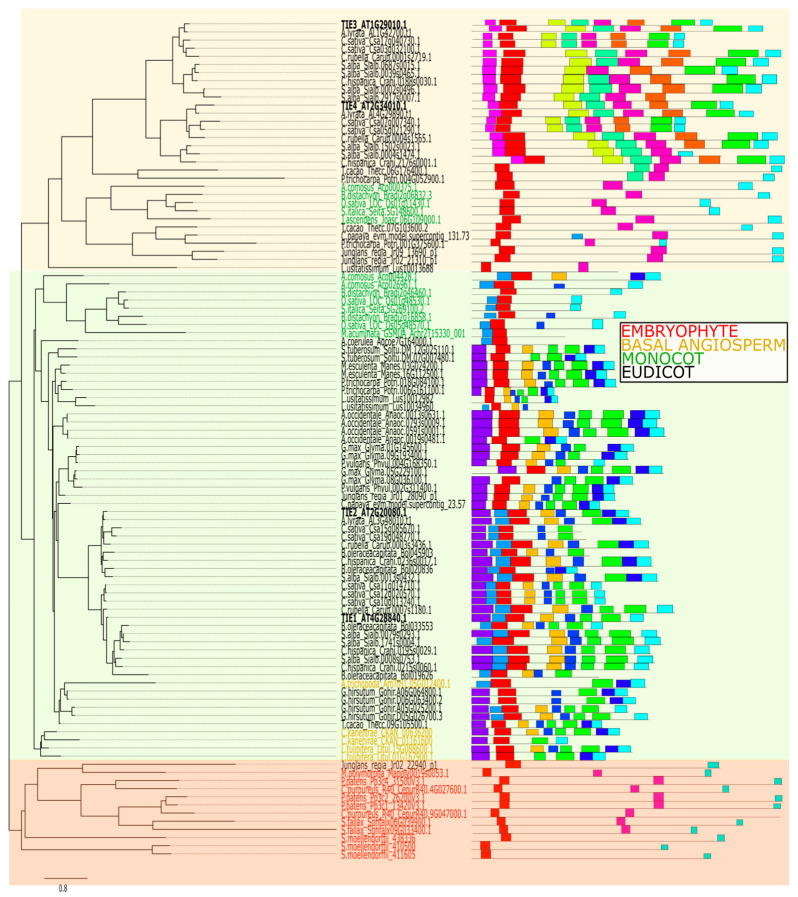
Protein motif analysis of the TIE transcriptional co-repressors family in embryophytes. Colored boxes represent individual motifs positioned next to each protein sequence on the tree structure from Figure 1 identified through the MEME suite. The specific sequence details and logos for each colored motif are provided in Figure 4, using the same color scheme for easy cross-referencing.

**Figure 4 plants-14-02423-f004:**
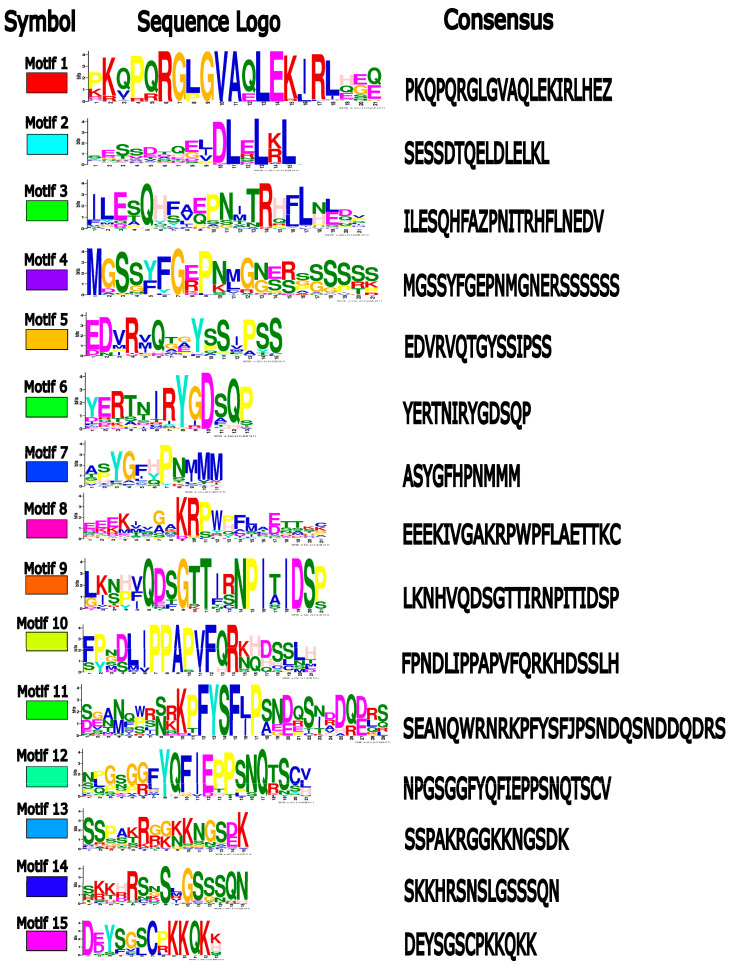
Protein motif consensus sequences of TIE proteins. Symbol, motif logo, and motif consensus sequence of the protein motifs depicted in Figure 3.

**Figure 5 plants-14-02423-f005:**
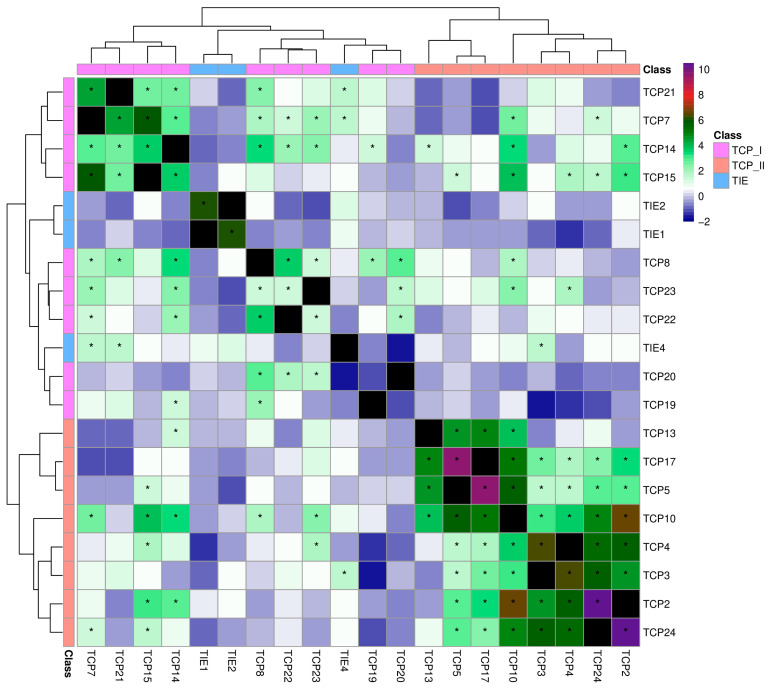
Co-expression profile of TIE and TCP genes in Arabidopsis. Heatmap of co-expression indices from the ATTED-II database for TIE and Class I and II TCP genes in Arabidopsis. Side colors indicate protein class. The correlation scores were obtained from ATTED-II (https://atted.jp/; accessed on 15 May 2025). Asterisk indicate co-expression z-scores above an alpha value of 0.1 in the normal distribution, indicating that they roughly correspond to the upper 10% of the most positively correlated co-expression gene pairs.

**Figure 6 plants-14-02423-f006:**
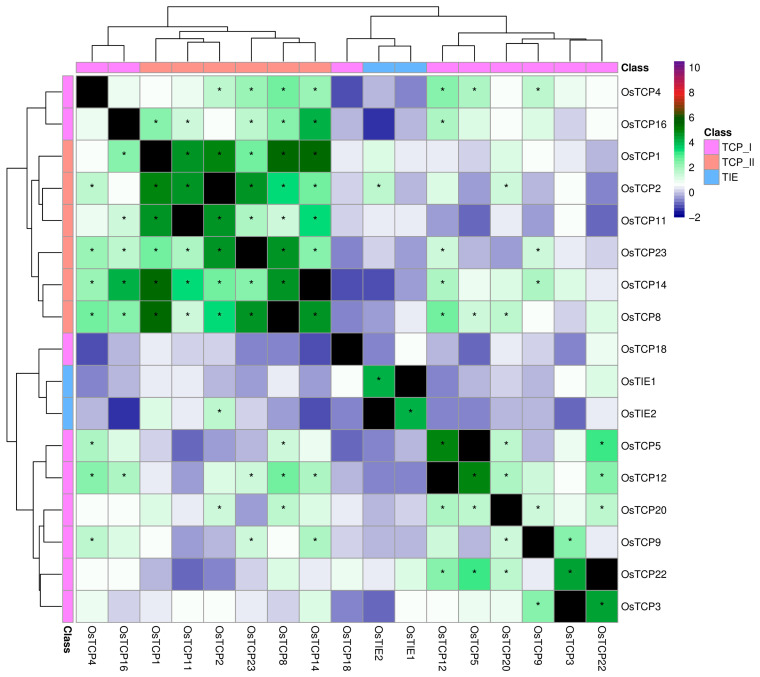
Co-expression profile of TIE and TCP genes in rice. Heatmap of co-expression indices from the ATTED-II database for TIE and Class I and II TCP genes in rice. Side colors indicate protein class. The correlation scores were obtained from ATTED-II (https://atted.jp/ accessed on 20 May 2025). Asterisk indicate co-expression z-scores above an alpha value of 0.1 in the normal distribution, indicating that they roughly correspond to the upper 10% of the most positively correlated co-expression gene pairs.

## Data Availability

Data are contained within the article.

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
