# Peer review of "Evolutionary Analysis of the Land Plant-Specific TCP Interactor Containing EAR Motif Protein (TIE) Family of Transcriptional Corepressors"

_plants, 2025, doi:10.3390/plants14152423_

Round 1

Reviewer 1 Report

Comments and Suggestions for Authors This study addresses the important issue establishing the origin of the TEI family of transcriptional corepressors involved in modulating the activity of TCP transcription factors. TCP TFs play an important role in regulation of plant development and several other aspects. In general, the work makes a good impression. It has been carried out using modern bioinformatics methods. A professional phylogenetic analysis was made. New motifs have been identified in TEI proteins. Using gene expression banks, co-expression genes form different group of TEIs and TCPs was revealed. However, as assumed by the title the research suggests a comprehensive evolutionary analysis of the TEI gene family. Thus would be better to see a more detailed study that includes other plant groups, hornworts, ferns and gymnosperms at least. The genomic information for these plant groups is available in specialized open databases. This will allow to more accurately determine the moment of emergence of phylogenetic clades that are characteristic of flowering plants. Gymnosperms are most interesting in this case. In addition, a more complete description of the absence of TEI representatives in algae would be appreciated, with particular attention to the Zygnematophyceae group, which is ancestral to land plants. And if TEIs are absent there, information about the group of organisms in which the closest relatives of TEIs are present would be useful to suggest parallel gene transfer. As a minor remark, it should be noted that in the results section the link to the algae database is indicated incorrectly.

Author Response

R1:

This study addresses the important issue establishing the origin of the TEI family of transcriptional corepressors involved in modulating the activity of TCP transcription factors. TCP TFs play an important role in regulation of plant development and several other aspects. In general, the work makes a good impression. It has been carried out using modern bioinformatics methods. A professional phylogenetic analysis was made. New motifs have been identified in TEI proteins. Using gene expression banks, co-expression genes form different group of TEIs and TCPs was revealed. However, as assumed by the title the research suggests a comprehensive evolutionary analysis of the TEI gene family. Thus would be better to see a more detailed study that includes other plant groups, hornworts, ferns and gymnosperms at least. The genomic information for these plant groups is available in specialized open databases. This will allow to more accurately determine the moment of emergence of phylogenetic clades that are characteristic of flowering plants. Gymnosperms are most interesting in this case. In addition, a more complete description of the absence of TEI representatives in algae would be appreciated, with particular attention to the Zygnematophyceae group, which is ancestral to land plants. And if TEIs are absent there, information about the group of organisms in which the closest relatives of TEIs are present would be useful to suggest parallel gene transfer. As a minor remark, it should be noted that in the results section the link to the algae database is indicated incorrectly.

R1 response:

We thank the reviewer for his/her assessment and encouraging feedback on our work. We have carefully considered all their suggestions to improve our manuscript.

As suggested, we conducted a thorough search for relevant sequences, including algae genomes from two public repositories: Mycocosm and Phytozome (see lines 90–94). Notably, the first portal includes genomes from the Zygnematophyceae group, but no hits were identified. Additionally, by consulting Phytozome and the Bryogenome portal, we ensured coverage across all Viridiplantae (land plants).

We agree with the reviewer on the importance of including basal plant groups. Our analysis already encompasses algae, mosses, hornworts, liverworts, and angiosperms. Gymnosperms were excluded because they have not yet been incorporated into the Phytozome portal due to insufficient genome sequence quality. To maintain rigor, we opted to use only well-curated genomes. Furthermore, since TIE sequences from basal angiosperm genomes exclusively group within Clade A (with strong support values), our phylogenetic analysis supports the origin of Clade B within angiosperms.

We have also corrected the link to the algae database as indicated (Mycocosm). We thank the reviewer for catching this error.

Reviewer 2 Report

Comments and Suggestions for Authors

In this study, the TCP Interactor containing EAR motif protein (TIE) was identified and characterized through phylogenetic analysis, gene co-expression analysis, and protein motif analysis. The results showed that the TIE family originated in the early evolution of embryophytes, with limited diversification in angiosperms compared to the expansion of TCP transcription factors. Co-expression analyses in Arabidopsis and rice revealed conserved expression patterns between TIE and TCP Class I genes, suggesting conserved regulatory mechanisms. However, the experimental design lacks rigor, and the Materials and Methods section is insufficiently detailed, requiring critical improvements.

Major point:

All results are in silico prediction. It is recommended authors conduct several experiments such as subcellular localization, RT-qPCR, transcriptional activity, and so on.

Minor points:

  1. In section 2.1, the authors only searched for algal TIE homologs through database mining, but did not mention whether experimental validation (e.g., RT-PCR) was performed to confirm the absence of TIE in algae. It is recommended to supplement RT-PCR experiments to confirm TIE absence.
  2. In section 2.4, the authors identified various co-expression relationships but did not validate protein interactions through experiments such as yeast two-hybrid or bimolecular fluorescence complementation, nor did they verify functions in mutants. It is suggested to Validate protein interactions and functional roles using transgenic or mutant plants (e.g., TIE-overexpressing lines or knockout mutants).
  3. The study only analyzes TIE origin and co-expression patterns but does not explore TIE function in stress resistance. It is suggested to conduct an expression profile analysis of the TIE genes in Arabidopsis under non-biological/biological stress conditions (such as salt, drought, and pathogen infection), and to verify their biological functions through overexpression experiments.
  4. In section 2.2, the authors propose that "TIE proteins may repress multiple TCP TFs" to explain their limited number. It is recommended to validate the selection pressure in plant evolution by combining Ka/Ks analysis.
  5. In the Materials and Methods section, please provide a detailed list of BLAST parameters (e.g., e-value < 1e-5), and list all the species included in the analysis. And specify the specific parameters for processing RNA-Seq transcript data.
  6. It is suggested to unify the format of subheadings, either all using noun phrases or all using complete sentences.
  7. Some Figures require modification:

All the pictures are not very beautiful; it is recommended to use professional software to beautify.

The text in Figure 1 is too crowded; adjust the font spacing.

In Figure 3, avoid excessive compression of graphics and legend text.

In Figures 4 and 5, the statistical significance (p-value) of the co-expression coefficient is not indicated, and the calculation method for the heatmap data should be supplemented.

Comments on the Quality of English Language

It is recommended to edit the language by native professor or language editing and proofreading company.

Author Response

R2:

In this study, the TCP Interactor containing EAR motif protein (TIE) was identified and characterized through phylogenetic analysis, gene co-expression analysis, and protein motif analysis. The results showed that the TIE family originated in the early evolution of embryophytes, with limited diversification in angiosperms compared to the expansion of TCP transcription factors. Co-expression analyses in Arabidopsis and rice revealed conserved expression patterns between TIE and TCP Class I genes, suggesting conserved regulatory mechanisms. However, the experimental design lacks rigor, and the Materials and Methods section is insufficiently detailed, requiring critical improvements.

We thank the reviewer for his/her assessment and encouraging feedback on our work. We have carefully considered all their suggestions to increase the rigor of our analyses and improve our manuscript accordingly. We responded point-by-point below.

Major point:

All results are in silico prediction. It is recommended authors conduct several experiments such as subcellular localization, RT-qPCR, transcriptional activity, and so on.

We understand the reviewer’s concern but the suggested experiments are out of the scope of this work, which is provide a phylogenetic framework to illuminate future functional experiments. Considering that the data used for co-expression analyses incorporate transcriptomic data from multiple RNA-seq experiments, these are real data from different tissue samples (not in silico). In addition, the use of genome sequence for phylogenetic analysis is the gold standard to study molecular evolution, which are not a prediction as well (is an inference to the past instead). To analyze sub-cellular localization we incorporate data from the literature.

Minor points:

  1. In section 2.1, the authors only searched for algal TIE homologs through database mining, but did not mention whether experimental validation (e.g., RT-PCR) was performed to confirm the absence of TIE in algae. It is recommended to supplement RT-PCR experiments to confirm TIE absence.

In order to perform RT-qPCR the experiment requires design specific primers. This task is impossible in absence of a gene sequence. In addition, lack of transcription not necessarily implies the absence of a gene. For example, if a specific gene is not expressed in the samples used for mRNA extraction. Therefore, RT-qPCR cannot be performed here for algae.

2. In section 2.4, the authors identified various co-expression relationships but did not validate protein interactions through experiments such as yeast two-hybrid or bimolecular fluorescence complementation, nor did they verify functions in mutants. It is suggested to Validate protein interactions and functional roles using transgenic or mutant plants (e.g., TIE-overexpressing lines or knockout mutants).

We agree about the importance of performing molecular and functional experiments to explore TIE protein function and evolution deeper. However, this is not the aim of this manuscript. On the other hand, those experiments will require a guiding hypothesis. We did discuss the known roles and molecular functions of TIE protein already reported. Interestingly, our co-expression analyses might illuminate future protein-protein interaction screenings. 

3. The study only analyzes TIE origin and co-expression patterns but does not explore TIE function in stress resistance. It is suggested to conduct an expression profile analysis of the TIE genes in Arabidopsis under non-biological/biological stress conditions (such as salt, drought, and pathogen infection), and to verify their biological functions through overexpression experiments.

Thanks for this suggestion. We agree that another conditions might give us valuable insights. Unfortunately, we have a limited time for revision of only two weeks. Therefore, in order to maintain rigorousness in our analyses as requested, the time is insufficient to include additional conditions into the expression analyses.  

4. In section 2.2, the authors propose that "TIE proteins may repress multiple TCP TFs" to explain their limited number. It is recommended to validate the selection pressure in plant evolution by combining Ka/Ks analysis.

This idea is really interesting for us to. We made an attemp to analyse Ka/Ks values, but due to the massive presence of disorganized regions on TIE sequences the results were probably biased, suggestion neutral evolution. There is no a clear way to deal with this problem in a more unbiased way. We decided not to include this data for lack of rigor.

5. In the Materials and Methods section, please provide a detailed list of BLAST parameters (e.g., e-value < 1e-5), and list all the species included in the analysis. And specify the specific parameters for processing RNA-Seq transcript data.

We provided the information as requested.

A deeper description of the parameters used for the Blast searches was necessary. To maximize the chances of finding a distant homolog we modified different Blast parameters. We chose the smallest word size available at NCBI, in order to allow for short seed matches that initiate an alignment. We changed the default substitution matrix to a matrix better suited for long evolutionary distances (as BLOSUM45 or PAM250). We increased the E-value threshold to include all potential hits, although given is not extremely short, we would not consider a hit homologous with E-values above 10E-6, as it is usually recommended. We also use nr database.

We performed different searches:

Query

Program

Database

Organism

Word size

Max E-value

Matrix

Best hit

Accession

Best hit E-value

Best hit

Percent Identity

Best hit

Query Cover

M.polymorpha

Mapoly0019s0053.1

blastp

nr

Chlorophyta (taxid: 3041, green algae)

2

10

BLOSUM45

KAK9864944.1

3.4

25.71

12%

Arabidopsis SPEAR3/TIE1 (Uniprot Q6IDB0)

blastp

nr

Chlorophyta (taxid: 3041, green algae)

2

10

BLOSUM45

KAF5832023.1

0.041

20.86

95%

As can be seen in the table, E-values are too high to consider any of the hits homologous. In relation to the search with the “TIE domain”, TIEs lack a defined domain, they possess a EAR motif, which is a short sequence, and disordered regions. Despite this, Blast performs local alignments, so cutting the protein is not required, if there is a partial alignment, it will be found. Cutting the protein might be interesting when searching for homologous proteins only to some regions/domain within the query and not to others, which is not our case.

6. It is suggested to unify the format of subheadings, either all using noun phrases or all using complete sentences.

We revised the subheading accordingly.

7. Some Figures require modification:

All the pictures are not very beautiful; it is recommended to use professional software to beautify.

We improved the definition and quality of the Figures as suggested.

8. The text in Figure 1 is too crowded; adjust the font spacing.

We increase the font size.  We were asked during the editorial assessment to incorporate complete sequence names and ID. This limits the space in a big phylogenetic tree.

9. In Figure 3, avoid excessive compression of graphics and legend text.

We divided Figure 3 into Figure 3 and Figure 4, to avoid compression.

10. In Figures 4 and 5, the statistical significance (p-value) of the co-expression coefficient is not indicated, and the calculation method for the heatmap data should be supplemented.

We incorporate this info to both figures (now Figure 5 and 6, respectively). In addition, we add information to the legends to better explain the data.

Reviewer 3 Report

Comments and Suggestions for Authors

The reviewed manuscript, titled "Comprehensive evolutionary analysis of the land plant-specific TCP Interactor containing EAR motif protein (TIE) Family of transcriptional corepressors” by Arce et al. is an incredibly brief communication that presents a relatively basic, straightforward results of a BLAST analysis of the TIE family of transcription factor interacting proteins.  Overall this is a very superficial work that really does not provide significant insight.

Major comments:

Section 2.1: the authors raise two potential scenarios that could be the origin of the TIE proteins.  Based on their BLAST analysis, they did not identify any hits in the algal lineages queried.  How thorough was the sampling done?  Id like to see the potential for a more reduced constraint on the search to see if/when potential hits begin to match and align.  Did the authors search for just the TIE domain itself – this might result in potential matches that may yield a better insight to help differentiate between their speculation and the data that is currently available.  Likewise, it appears from the methods that the authors searched the algal lineages for a match to the liverwort sequence – is there any reason not to expand this search as well?  Without an expansion and thorough study, the title of the article would best be limited in its scope.

Figures 1,2,3: each of these phylogeny images is incredibly poorly formatted.  The text overlaps and obscures the ability to clearly read and follow what the authors are trying to present.  The choice of yellow font on yellow background is particularly difficult to see and read.

Figure 3 presents a few distinct logo plots of conserved AA domains within this family.  Have the authors performed a protein BLAST for these?  Is there anything that has been identified before?  Are any of these 15 motifs characterized in any way?  This presentation gives the appearance that all of these motifs are novel, and yet there is very little discussion of them, functional information, or follow up.

Figure 4: the heatmap is not clearly presented or labelled in the legend for the image or the figure legend.

The transcriptional correlation analysis is challenging to reconcile with the author’s conclusions.  As far as the color scheme is concerned based on the key and the protein classification, it looks very much like there is an antagonistic/anticorrelation for the TIE and 2 TCP protein families overall.  Their hierarchal clustering does position the grouping, but their discussion is incredibly superficial overall.

Minor comments:

Line 20: check to make sure that proper formatting for genus and species names is consistent throughout – e.g. Arabidopsis is not italicized.

Author Response

R#3

The reviewed manuscript, titled "Comprehensive evolutionary analysis of the land plant-specific TCP Interactor containing EAR motif protein (TIE) Family of transcriptional corepressors” by Arce et al. is an incredibly brief communication that presents a relatively basic, straightforward results of a BLAST analysis of the TIE family of transcription factor interacting proteins.  Overall this is a very superficial work that really does not provide significant insight.

 Major comments:

Section 2.1: the authors raise two potential scenarios that could be the origin of the TIE proteins.  Based on their BLAST analysis, they did not identify any hits in the algal lineages queried.  How thorough was the sampling done?  Id like to see the potential for a more reduced constraint on the search to see if/when potential hits begin to match and align.  Did the authors search for just the TIE domain itself – this might result in potential matches that may yield a better insight to help differentiate between their speculation and the data that is currently available.  Likewise, it appears from the methods that the authors searched the algal lineages for a match to the liverwort sequence – is there any reason not to expand this search as well?  Without an expansion and thorough study, the title of the article would best be limited in its scope.    

Thank you for your suggestions, given the relevance of this finding for our conclusions, a deeper description of the parameters used for the Blast searches was necessary. To maximize the chances of finding a distant homolog we modified different Blast parameters. We chose the smallest word size available at NCBI, in order to allow for short seed matches that initiate an alignment. We changed the default substitution matrix to a matrix better suited for long evolutionary distances (as BLOSUM45 or PAM250). We increased the E-value threshold to include all potential hits, although given is not extremely short, we would not consider a hit homologous with E-values above 10E-6, as it is usually recommended. We also use nr database.

We performed different searches:

Query

Program

Database

Organism

Word size

Max E-value

Matrix

Best hit

Accession

Best hit E-value

Best hit

Percent Identity

Best hit

Query Cover

M.polymorpha

Mapoly0019s0053.1

blastp

nr

Chlorophyta (taxid: 3041, green algae)

2

10

BLOSUM45

KAK9864944.1

3.4

25.71

12%

Arabidopsis SPEAR3/TIE1 (Uniprot Q6IDB0)

blastp

nr

Chlorophyta (taxid: 3041, green algae)

2

10

BLOSUM45

KAF5832023.1

0.041

20.86

95%

As can be seen in the table, E-values are too high to consider any of the hits homologous. In relation to the search with the “TIE domain”, TIEs lack a defined domain, they possess a EAR motif, which is a short sequence, and disordered regions. Despite this, Blast performs local alignments, so cutting the protein is not required, if there is a partial alignment, it will be found. Cutting the protein might be interesting when searching for homologous proteins only to some regions/domain within the query and not to others, which is not our case.

Figures 1,2,3: each of these phylogeny images is incredibly poorly formatted.  The text overlaps and obscures the ability to clearly read and follow what the authors are trying to present.  The choice of yellow font on yellow background is particularly difficult to see and read.

We improve the quality of the images for chariness as suggested. We have increased the font size and final resolution of the files.

Figure 3 presents a few distinct logo plots of conserved AA domains within this family.  Have the authors performed a protein BLAST for these?  Is there anything that has been identified before?  Are any of these 15 motifs characterized in any way?  This presentation gives the appearance that all of these motifs are novel, and yet there is very little discussion of them, functional information, or follow up.

We already mentioned that motif 1 and 2 contains known motifs. We now further discuss this in lines 167-188 to better portrait their importance. The other motifs here found were not previously characterized. 

Figure 4: the heatmap is not clearly presented or labelled in the legend for the image or the figure legend.

The transcriptional correlation analysis is challenging to reconcile with the author’s conclusions.  As far as the color scheme is concerned based on the key and the protein classification, it looks very much like there is an antagonistic/anticorrelation for the TIE and 2 TCP protein families overall.  Their hierarchal clustering does position the grouping, but their discussion is incredibly superficial overall.

Our analyses identified a previously unknown correlation between TIE and TCP proteins, revealing distinctiveness for Class I and Class II TCPs.  This outcome is of interest of plant biologist focused in TCPs and/or TIE proteins as well. Moreover, we found that this correlation is similar in two distant related plant species, which highlights the need and importance of the phylogenetic framework here provided.

Minor comments:

Line 20: check to make sure that proper formatting for genus and species names is consistent throughout – e.g. Arabidopsis is not italicized.

Checked.

Round 2

Reviewer 1 Report

Comments and Suggestions for Authors

I thank the authors for their detailed answers to my comments In general, they seem quite convincing and wellfounded. However, I did not receive an answer to my last question about the parallel transfer of genes from other organisms. I believe that gene groups cannot appear out of nowhere. Relatives of genes may exist in some groups of living organisms. Therefore, I would like tto ask he authors to conduct a wider search for the possible relatives of this gene group using GenBank, for example.

Author Response

R#1

I thank the authors for their detailed answers to my comments. In general, they seem quite convincing and well founded. However, I did not receive an answer to my last question about the parallel transfer of genes from other organisms. I believe that gene groups cannot appear out of nowhere. Relatives of genes may exist in some groups of living organisms. Therefore, I would like tto ask he authors to conduct a wider search for the possible relatives of this gene group using GenBank, for example.

Reply:

We appreciate the reviewer’s positive feedback on our manuscript. We have considered this concern carefully and have tried to the best of our ability to address the question regarding a potential horizontal gene transfer (HGT) origin for TIE.

Our observation that TIE proteins are absent from the algal genomes we searched does not imply that TIE genes evolved de novo without an ancestral gene. However, the inability of Blast searches to identify a homologous gene suggests that sequence conservation supporting homology may have been eroded by mutations during evolution. This erosion could potentially result from the neofunctionalization of a "proto-TIE" gene, leading to new selective pressures acting on its sequence and ultimately giving rise to the TIE genes found in present-day plants. Furthermore, the lack of strongly conserved domains and the presence of disordered regions may reduce the statistical power of algorithms to identify very distant homologs.

To further investigate a potential HGT origin, we searched for homologs of TIE outside Viridiplantae using M. polymorpha and A. thaliana TIE proteins as queries against the non-redundant database. Again, no significant hits were identified. Consequently, we are unable to propose an HGT event as the origin of the TIE protein in plants.

Query

Program

Database

Organism

Word size

Max E-value

Matrix

Best hit

Accession

Best hit E-value

Best hit

Percent Identity

Best hit

Query Cover

M.polymorpha

Mapoly0019s0053.1

blastp

nr

Exclude green plants (Viridiplantae,taxid:33090)

2

1

BLOSUM45

No hit

-

-

-

Arabidopsis TIE1 (Uniprot Q6IDB0)

blastp

nr

Exclude green plants (Viridiplantae,taxid:33090)

2

1

BLOSUM45

No hit

-

-

-

Reviewer 2 Report

Comments and Suggestions for Authors

OK

Author Response

We appreciate the Reviewer's assessment of our work.

Reviewer 3 Report

Comments and Suggestions for Authors

I approve of the changes and the authors responses.

Author Response

(The authors gave the same response as above.)
